# Chromium Complexes Supported by Salen-Type Ligands for the Synthesis of Polyesters, Polycarbonates, and Their Copolymers through Chemoselective Catalysis

**DOI:** 10.3390/ijms24087642

**Published:** 2023-04-21

**Authors:** Ilaria Grimaldi, Federica Santulli, Marina Lamberti, Mina Mazzeo

**Affiliations:** Department of Chemistry and Biology “Adolfo Zambelli”, University of Salerno, Via Giovanni Paolo II, 132, 84084 Fisciano, SA, Italy; ilagrimaldi@unisa.it (I.G.); fsantulli@unisa.it (F.S.)

**Keywords:** block copolymers, CO_2_ fixation, chromium (III) catalysts, depolymerization, polycarbonates, polyesters

## Abstract

Salen, Salan, and Salalen chromium (III) chloride complexes have been investigated as catalysts for the ring-opening copolymerization reactions of cyclohexene oxide (CHO) with CO_2_ and of phthalic anhydride (PA) with limonene oxide (LO) or cyclohexene oxide (CHO). In the production of polycarbonates, the more flexible skeleton of salalen and salan ancillary ligands favors high activity. Differently, in the copolymerization of phthalic anhydride with the epoxides, the salen complex showed the best performance. Diblock polycarbonate-polyester copolymers were selectively obtained by one-pot procedures from mixtures of CO_2_, cyclohexene oxide, and phthalic anhydride with all complexes. In addition, all chromium complexes were revealed to be very active in the chemical depolymerization of polycyclohexene carbonate producing cyclohexene oxide with high selectivity, thus offering the opportunity to close the loop on the life of these materials.

## 1. Introduction

The current concerns about plastic pollution and the depletion of oil reserves are driving scientific research toward the development of more sustainable polymers [1,2]. In this context, aliphatic polyesters and aliphatic polycarbonates represent very promising alternatives to traditional plastics because of their general biocompatibility and agile hydrolytic degradation [3]. The ring-opening copolymerizations (ROCOP) of epoxides with CO_2_ or cyclic anhydrides are versatile methods for the synthesis of a wide range of polycarbonates [4] and polyesters [4,5].

Compared to the ring-opening polymerization (ROP) of lactones, the ROCOP of cyclic anhydrides and epoxides provides access to more structurally diverse polyesters, both aliphatic and semi-aromatic, thanks to the large variety of low-cost and commercially available monomers, in some cases obtained from renewable resources [6].

On its side, the ring-opening copolymerization of epoxides and CO_2_ allows the production of biodegradable and biocompatible polycarbonates, exploiting the CO_2_ as an abundant, inexpensive, safe, and renewable C1 source ensuring, at the same time, its fixation in durable materials [7,8,9,10,11].

New opportunities to amply the availability of advanced materials with hybrid and tunable properties are offered by the synthesis of polycarbonate-polyester block copolymers that conjugate the chemical properties of the related homopolymers into a single structure [12,13]. Generally, polycarbonate-polyester block copolymers are synthesized by sequential polymerization reactions and/or with macro-initiators. These methods require multistep processes in which tandem or multi-functional catalysis is involved, thus lacking extensive versatility [14].

In 2014, Williams reported the first example of “switch catalysis” for the selective production of diblock polycarbonate-polyester copolymers obtained from a mixture of caprolactone, cyclohexene oxide, and carbon dioxide by using a single homogeneous di-zinc catalyst [15]. Both kinetic and thermodynamic control by the metal-chain end group revealed to be responsible for the selectivity of the process in which the catalyst switches from a polymerization cycle to another depending on the nature of the propagating species. Subsequently, Rieger and co-workers obtained diblock/random copolymers from β-butyrolactone, cyclohexene oxide, and carbon dioxide by using a di-zinc β-diiminate catalyst [16].

In the last few years, Williams and other authors demonstrated the generality of switch catalysis and its applicability beyond zinc complexes, and interesting results have been obtained with catalysts based on different metals [13,17,18], such as chromium [19] and aluminum [20,21,22]. Among the catalysts studied for both reactions, as well as their combination, metal complexes bearing salen-type ligands have demonstrated very good performances in terms of activity and selectivity [23,24,25,26]. In particular, salen-based chromium complexes, in combination with anions derived from PPN^+^ salts, are among the most effective and robust catalysts for the ring-opening copolymerization of CO_2_ with epoxides [27,28]. Salen derivatives such as salan and salalen have also been tested as ancillary ligands for chromium complexes as catalysts in this class of reactions [7]. In the copolymerization of cyclohexene oxide (CHO) and CO_2_, salanCr complexes [29] showed a higher activity with respect to their salen counterparts by using PPNN_3_ as a co-catalyst. Later on, half-reduced salalen complexes in combination with PPNCl showed excellent activity in the production of PCHC, even under mild conditions [30]. In both cases, the high activity was attributed to the higher coordination flexibility of the salan and salalen systems with respect to the rigid salen-based complexes, resulting in a higher tendency to form metal complexes where nitrogen and oxygen donor atoms occupy three equatorial and one axial coordination sites [31].

In the copolymerization of epoxides with cyclic anhydrides, salen chromium catalysts have been largely investigated, showing very good performances [32,33,34,35]. A substantial influence of ligand structure was highlighted by several authors [36,37,38,39]. Differently, the related salan and salalen derivatives have been rarely explored in these reactions [40,41]. In the switch catalysis from monomer mixtures comprising epoxide, anhydride, and lactone or CO_2_, ABA triblock polyesters have been synthesized using a commercially available chromium catalyst [19,42] and sophisticated binuclear chromium complexes [37,38].

Considering that chromium complexes are among the catalysts that show the best performances in the cited catalysis, and well aware that even subtle modifications of several parameters (such as ligand substitutions, labile ligands on the metal, co-catalysts, reaction conditions, etc.) may cause dramatic influences on activity and/or selectivity of the catalytic systems [43], we decided to synthesize salen, salan, and salalen ligands with the same substituents on the phenolate moieties and the same bridge between the nitrogen atoms [44]. Then the corresponding chromium chloride complexes were prepared following the same procedure [31] and tested as catalysts for the ROCOP of CHO with CO_2_ under the same conditions. Subsequently, we explored the behavior of the chromium complexes in the copolymerization of phthalic anhydride (PA) with two different epoxides, CHO and limonene oxide (LO), for the synthesis of polyesters. Moreover, diblock polyester/polycarbonate polymers were prepared by chemoselective polymerization of CHO, CO_2_, and PA. For each polymerization reaction, the commercial salCyCrCl complex **4** was tested for comparison under the same reaction conditions. Finally, the synthesized chromium complexes have been used to promote the depolymerization of the polycarbonates, thus highlighting the degradability of these materials and, at the same time, the versatility of these catalysts.

## 2. Results and Discussion

### 2.1. Synthesis and Charactherization of Chromium Complexes

Chromium complexes **1**–**3** were synthesized according to a previously reported procedure [31]. The opportune proligand and one equivalent of chromium (II) chloride CrCl_2_ were dissolved in THF, and the solution was stirred at room temperature in a nitrogen atmosphere for 24 h (Figure 1). After this time, the solution was exposed to the air for two hours to allow the oxidation of chromium (II) to chromium (III). The desired complexes were obtained by extraction with diethyl and a saturated solution of NaCl. The organic phase was dried with Na_2_SO_4_, filtered, and the solvent was removed in vacuo. In Figure 2, the structures of complexes **1**–**3** are detailed, and the structure of the commercial complex **4** is also reported for comparison.

Complexes **1** [45,46], **2** [30], and **3** [29] have been previously reported in the literature, although a full characterization has never been described. The paramagnetic complexes were firstly analyzed by matrix-assisted laser desorption/ionization time-of-flight mass spectral (MALDI-ToF MS) (Appendix A).

For the salen- and salan-chromium complexes, the most intense peak was attributable to the species formed after the loss of one chloride. Interestingly, for the salalen-chromium complex, the higher peak corresponded to the species formed after the loss of the chloride and the coordination of one oxygen molecule [47].

Complexes **1**–**3** were then analyzed by UV-visible spectroscopy, and their spectra were compared to those of the free ligands (Figure 1). Samples were dissolved in acetonitrile (80 µmol), and the absorbance was recorded from 200 to 800 nm (Appendix A). Indicative of the formation of the desired chromium complexes, phenolate ligand-to-metal charge-transfer (LMCT) transitions were observed; these bands appeared at lower energies and with lower intensities with respect to the bands also found in the spectra of the corresponding free ligands and attributable to the π → π* transitions of the phenolic chromophores. In particular, the LMCT bands were observed at 416, 432, and 569 nm, respectively, for complexes salalenCrCl **2**, salenCrCl **1,** and salanCrCl **3**.

The accomplishment of the ligands coordination to the chromium metal was further substantiated by IR characterization (IR spectra of ligands and complexes **1**–**3** are reported in Appendix A). In fact, in the spectra of each complex, there appeared two peaks around 550 and 490 cm^−1^, indicative of the M–N and M–O bonds, respectively.

Finally, the magnetic susceptibility data for complexes **1**–**3** were determined at room temperature by the Evans NMR method by using C_6_D_6_ as a solvent and cyclohexane as a diamagnetic reference. The magnetic moments of the complexes were in the range of 3.06–3.15 and thus close to the spin-only value of 3.99 μB for S = 3/2, indicating the formation of high-spin Cr(III) complexes.

### 2.2. Copolymerizations of CO_2_ with Cyclohexene Oxide (CHO)

First, we explored the behavior of complexes **1**–**4,** in combination with PPNCl as cocatalyst, in the copolymerization of CO_2_ with CHO (Figure 3), performed at a cyclohexene oxide/catalyst/cocatalyst ratio of 1000:1:1, under 13 bar of CO_2_ pressure and at a temperature of 70 °C without using any solvent (Table 1 and Appendix A).

As depicted in Figure 4, the copolymerization initiates with the activation of the epoxide by coordination to the Lewis acidic metal center, and the following nucleophilic attack of the initiating species X (e.g., the chloride ion from PPNCl) ring-open the monomer. The resulting metal alkoxide inserts a CO_2_ molecule (pathway b in Figure 4), generating a metal carbonate. Therefore, the cycle propagates by alternate repetitions of CHO and CO_2_ insertions, leading to the formation of the polycarbonate products.

For a coherent comparison, substrates with the same degree of purity and the same equipment were used for all experiments.

After 3 h, the product mixtures were analyzed by ^1^H NMR spectroscopy, with the quantities of polycyclohexene carbonate (PCHC), ether linkages in the PCHC, and CHO determined by integrating the resonances at 4.60, 3.45, and 3.11 ppm, respectively (see Appendix A). Worth noting, negligible quantities of ether linkages (<0.3%) were observed in each sample, highlighting the formation of completely alternating copolymers of CHO and CO_2_ with no homopolymerization of CHO. In addition, chromium complexes **1**–**3** result in high selectivity towards the formation of the polymeric product; in fact, no signals due to cyclic carbonates (either *cis*- or *trans*-cyclohexene carbonates) were detected in any case.

The conversion values indicated the order of activity: salalenCrCl **2** > salanCrCl **3** > salenCrCl **1** (compare entries 1–3 in Table 1), thus corroborating the previously stated notion that the greater flexibility of (half-) reduced ligands favors the activity of the corresponding complexes. In addition, the comparison of the two salen-based complexes **1** and **4** (compare entries 1 and 4, respectively, in Table 1) suggested that the cyclohexyl backbone favors the activity of the catalytic system with respect to the ethylene bridge.

The molecular weights of the copolymers were determined by gel permeation chromatography (GPC) in THF solution using a RI detector and polystyrene standards. Peak profiles of GPC of the resulting copolymers were bimodal, and the number-average molecular weights of the copolymers were smaller than the theoretical values; this is quite common in this kind of polymerization reactions, and it is generally ascribed to the occurrence of chain-transfer reactions promoted by contaminant water or attributable to the presence of cyclohexan-1,2-diol (CHD, produced by hydrolysis of CHO). However, molecular-weight distributions were relatively narrow (Ɖ = 1.17 ÷ 1.25), suggesting that the chain-transfer reactions were reversible and fast [48].

In addition to the main signals for the protons of the PCHC chains, the ^1^H NMR spectra of the polymer samples showed two peaks at 3.60 and 4.45 ppm, indicative of the formation of the cyclohexanol end group, which can be generated by hydrolysis in the termination stage of the polymerization.

The structural information of the obtained PCHC samples was also obtained by MALDI-ToF high-resolution MS. In particular, the MALDI-ToF spectrum of the sample obtained in entry 3 showed the presence of one major distribution separated by 142.06 Da, which is the mass of the polymer repeating unit, and centered around 6100 Da, in good agreement with molecular weights obtained by GPC (see Appendix A). The peak appearing at *m*/*z* 5956 can be closely matched with the calculated molecular weight of 40 repeating units of PCHC possessing a chloride group at the α-chain end and a hydroxyl group at the ω-chain end, suggesting the initiation reaction from a chloride (reasonably deriving from the initiator, PPNCl, see Figure 4), and supporting the termination reaction by hydrolysis, as already suggested by the NMR analysis.

The signals observed in the carbonate region of the ^13^C NMR spectrum of the PCHC copolymer obtained by complex **4**/PPNCl indicated an atactic structure despite the chiral nature of the complex. This result was previously reported by Darensbourg working with complex **4** and N-methylimidazole as cocatalyst [49], under different reaction conditions, suggesting that the ability of this complex to exert stereocontrol does not depend on the reaction conditions.

DSC analysis of polycarbonates obtained in runs 1–4 allowed us to identify their glass transition temperature (T_g_). The DSC thermograms (see Appendix A for a representative example) show T_g_ values of 120.92 °C, 111.11 °C, and 121.83 °C for entries 2, 3, and 4, respectively, in line with the values reported in the literature for completely alternating PCHC samples with different molecular weights [43].

Subsequently, the thermal decomposition of PCHC samples was studied by thermal gravimetric analysis (TGA). For PCHC samples reported in Table 1, the onset decomposition temperatures T_d_ (defined by the temperature of 5% weight loss) were in the range 219–248 °C (see Appendix A for a representative example).

### 2.3. Copolymerization of Cyclohexene Oxide (CHO) and Limonene Oxide (LO) with Phthalic Anhydride (PA)

Subsequently, catalysts **1**–**4**, in combination with the cocatalyst PPNCl, were tested for the copolymerization of phthalic anhydride (PA) with cyclohexene oxide (CHO) or limonene oxide (LO) (Figure 5).

Both PA and CHO are commercial chemicals used at scale in the polymer industry. Currently, they are produced from petroleum derivatives; however, alternative synthetic strategies from biomass have been individuated: CHO can be obtained from 1,4-cyclohexadiene, a waste product of plant oil self-metathesis [50], and PA from agricultural residue [51].

Differently (R)-limonene oxide (LO) is a bioderived monomer, naturally occurring as a mixture of *cis* and *trans* isomers, which is extracted from waste citrus fruit peel [52], thus polymers obtained from LO are bioderived materials [53,54,55,56].

All PA/CHO polymerizations were performed at 70 °C with a single equivalent of PPNCl and in the presence of an excess of CHO to avoid the use of solvent (Table 2 and Appendix A). The initial molar ratios of the comonomers and the complexes were [PA]_0_:[CHO]_0_:[Cr]_0_:[PPNCl]_0_ = 1000:100:1:1 when using CHO as an epoxide, and [PA]_0_:[LO]_0_:[Cr]_0_:[PPNCl]_0_ = 1000:500:1:1 for the copolymerization with LO.

The mechanism of the ROCOP of epoxides with anhydrides is similar to that previously described for the ROCOP of epoxides with CO_2_. The initiation step consists in the nucleophilic attack by the initiator on a CHO unit followed by the insertion of the anhydride to generate the related carboxilate derivative. The alternate insertions of the two monomers allow the formation of the polyester chain (Figure 4, pathway a).

The monitoring of the conversion of the phthalic anhydride versus time and the selectivity of the reaction (expressed as the content of ester linkages in the polymer chain) were evaluated by ^1^H-NMR spectroscopy. The molar masses of the obtained polyesters and their dispersities were determined by GPC analysis using polystyrene standards.

In the ROCOP of PA with CHO, the Salen-Cr complex (**1**) revealed itself to be the most active catalyst, showing, after 30 min, a conversion almost quantitative (93%) of PA (entry 1, Table 2). After the same reaction time, significantly lower reactivities were observed with the Salalen-Cr and Salan-Cr complexes (46% and 31%, respectively; entries 2 and 3, Appendix A); however, for both complexes, almost quantitative conversions were obtained only after 1 h (entries 2 and 3, Table 2).

The activity of the Salen-Cr complex was comparable to that reported for the Salen-Cr bearing a cyclohexyl−diimine backbone [39] (complex **4**), although more drastic reaction conditions were used (T = 130 °C), and much higher than that of the salophen chromium (III) complex, for which a conversion of PA of around 4% was achieved after 60 min [34]. In both cases, the complexes were activated by DMAP (4-dimethylaminopyridine).

For a more coherent comparison, a polymerization test was performed with **4** in the presence of PPNCl, obtaining a conversion analogous to that of complex **1** (compare entries 4 and 1, Table 2). This is evidence that the nature of the cocatalyst has a more significant influence on the catalytic activity than the backbone skeleton of the ancillary ligand.

In the copolymerization of PA with LO, the same trend of activity was observed with the salen complex, which showed the highest activity, allowing the conversion of 63% of the phthalic anhydride after 15 min (entries 5–7, Table 2 see also Appendix A). These activities were higher than those observed with salen aluminum complexes [57] or with a salophen chromium complex [56]. However, the best performances were obtained with the commercial chromium complex **4** (entry 8, Table 2).

Both copolymerizations proceeded in an exclusive alternating fashion with ester linkage content > 99%, as evident by the ^1^H NMR spectra of polycyclohexene phthalate and polylimonene phthalate that showed perfectly alternated microstructures in which no polyether sequences were detected (Appendix A).

The average molecular weight values (M_n_) measured by GPC (without any calibration correction) were lower than the theoretical ones expected for a living system. This result is frequently observed in this catalysis as a consequence of the presence of protic impurity traces (small amounts of hydrolyzed anhydrides or cyclohexene oxide) that can act as chain transfer agents [34,58,59], as previously discussed for the copolymerization of CHO with CO_2_. These copolymerization processes are susceptible to rapid displacement of the growing polymer chain by protic species; thus, the new metal-alkoxide or hydroxide bond formed can initiate a new polymer chain, leading to shorter molecular weight polymers.

This behavior could also explain the profile of the GPC curves, in which a peak related to a second and less abundant distribution is evident (Appendix A).

The glass transition temperatures fall in the range between 122 °C and 104.9 °C, values coherent with those reported in the literature for alternating CHO/PA and LO/PA polyesters, respectively (Appendix A) [34]. The thermal decompositions of polyesters were studied by TGA (Appendix A).

### 2.4. Terpolymerization of Cyclohexene Oxide (CHO) with Phthalic Anhydride (PA) and CO_2_

Once the activities of all the systems in the two different ROCOPs were evaluated, we tested their performance in the synthesis of polyesters/polycarbonates diblock by combining the two processes (Figure 4 and Figure 6).

The reactions were conducted at 70 °C with a CHO:PA:cat:PPNCl ratio of 1000:100:1:1 under 13 bar of CO_2_ pressure. The reaction conditions were analogous to those used for the related copolymerizations. Representative results are described in Table 3 and Appendix A. After 3 h, an aliquot of the reaction mixture for each terpolymerization was examined by ^1^H NMR spectroscopy, revealing, in all cases, the complete consumption of the PA. This is coherent with that observed by Darensbourg for the terpolymerization of CO_2_/PA/CHO promoted by complex **4,** in which the formation of the polycarbonate block occurs only after the complete consumption of the anhydride (see Figure 4) [42].

The microstructures of copolymers were examined by ^13^C{^1^H} NMR spectroscopy, which showed only signals relative to the two blocks with no additional signals, and by DOSY NMR spectra (Figure 2 and Appendix A) showing a single diffusion coefficient for all signals (entry 2 of Table 3). Both data are consistent with the formation of diblock polyester/polycarbonate copolymers.

The ^1^H NMR spectra allowed us to calculate the relative lengths of the two blocks that were consistent with the activities observed for the four catalysts in the two copolymerization processes. The ^1^H NMR spectrum of an isolated diblock coplymer is reported as a representative example in Figure 3 and Appendix A.

The GPC analysis of the isolated copolymers (Appendix A) exhibited M_n_ values consistent with the sum of contributions related to the two component blocks. The distributions showed the same shape observed for the related PCHC and narrow dispersities Đ < 1.25.

In the MALDI-TOF spectra of the diblock copolymers, different distributions were evident. For the sample of entry 2 of Table 3, in the first series, centered at 4500 *m*/*z*, two families of peaks were detected corresponding to Cl or OH end-capped chains in which CHO-PA sequences are directly joined to PCHC sequences. A distribution centered at 8000 *m*/*z* described di-block structures in which perfectly alternated CHO-PA units were joined to sequences of PCHC by isolated cyclohexene oxide units (Figure 4).

The thermal performances of the obtained poly-(ester-b-carbonate)s were studied by DSC and thermogravimetric analysis (TGA). These exhibit glass transition temperatures (Tg) around 116 °C (Appendix A), a value a little lower than the T_g_ of PCHC, and good thermal stability with a T_d_ (5%) > 230 °C (Appendix A).

### 2.5. Chemical Depolymerization of PCHC

Chemically recyclable polymers that degrade into the corresponding monomers or other chemicals are emerging as strategic materials to reduce the impact of plastic pollution on the environment [60,61]. Recently, about the depolymerization of polycarbonates [62,63], Lu, Liu, and coworkers reported the depolymerization of PCHC promoted by metal (III) complexes bearing salen-based ligands [64]. Among them, the commercial chromium complex **4**, in combination with PPNN_3_, exhibited high efficiency and selectivity in the depolymerization of different polycarbonates.

In light of these intriguing results, in this work we decided to verify the ability of chromium complexes **1**–**3** to promote the depolymerization reactions of polycyclohexene carbonates (Table 4 and Appendix A).

The most abundant PCHC sample (obtained in entry 2 of Table 1), whose thermal stability had been evaluated by TGA analysis (T_d_ = 248 °C), was used to compare the performance of the synthesized complexes **1**–**3** in the depolymerization reactions (see Figure 7), working under conditions similar to those explored by Lu [64]. Operating in bulk at 190 °C with 0.2 mol % catalyst loading and one equivalent of PPNCl (entry 1, Table 4), salanCrCl complex **1** promoted the almost full depolymerization of the PCHC sample (94% of conversion) with a selectivity of 97% versus the monomer (CHO) in only ten minutes.

Thus, we reduced the depolymerization time to five minutes and compared the behavior of complexes **1**–**3** under the same reaction conditions (entries 2–4, Table 4). To our delight, all three complexes depolymerize the PCHC sample in CHO with high selectivity (95–96%), with the salen complex **1** showing the highest activity (75% of conversion), followed by salalen complex **2** (56%), and then by the salan complex **3** (12%).

The observed differences among the three catalysts are quite significant and, at the same time, quite unexpected, considering that subtle variations of the complex structures do not tend to play a role in catalyst activities when operating under such harsh conditions.

Finally, a depolymerization test of the poly-(ester-b-carbonate) was also performed (entry 5, Table 4). After five minutes, a depolymerization of 12% of the polycarbonate block was observed, while the polyester portion remained intact.

In their paper, Lu and collaborators suggest a two-step process in which the polymer is first converted into *trans*-CHC and then subsequently into CHO, assuming that the catalyst plays a role in both steps; however, no type of interaction between the metal and the polymeric chain is proposed. Thus, our interesting results definitely need more in-depth mechanistic studies, which will certainly be conducted in due course.

## 3. Materials and Methods

### 3.1. Reagents and Methods

All the operations of synthesis and handling of air-sensitive chemicals were performed in an inert atmosphere, using Schlenk techniques and/or a glove box in a nitrogen atmosphere. The used glassware was dried in an oven at 120  °C and subsequently subjected to vacuum-nitrogen cycles. Benzene, hexane, and toluene (Sigma-Aldrich, St. Louis, MO, USA) were distilled under nitrogen over sodium benzophenone. Deuterated solvents were purchased from Sigma-Aldrich and dried over activated 3-Å molecular sieves prior to use. All the reagents used for the synthesis of the complexes and the lactide were purchased from Sigma-Aldrich.

### 3.2. Instruments

The NMR spectra were recorded with BRUKER ADVANCE instruments operating at 600, 400, and 300 MHz for ^1^H. Chemical shifts (δ) are expressed in parts per million and coupling constants (J) in hertz. ^1^H NMR spectra are referenced using the residual solvent peak, δ = 7.27 for CDCl_3_. _13_C NMR spectra are referenced using the residual solvent peak at δ = 77.23 for CDCl_3_. The measurement of diffusion has been carried out by observing the attenuation of the NMR signals during a pulsed field gradient experiment using the double-stimulated echo pulse sequence. In particular, 2D DOSY PGSE NMR spectra were performed on a Bruker Avance 600 spectrometer at 298 K without spinning.

The molecular masses (M_n_ and M_w_) and their dispersities (M_w_/M_n_) were measured by gel permeation chromatography (GPC), using THF as the eluent (1.0 mL min^−1^), and narrow polystyrene standards were used as the reference.

MALDI mass spectra were recorded using a Bruker solariX XR Fourier transform ion cyclotron resonance (FT-ICR) mass spectrometer (Bruker Daltonik GmbH, Bremen, Germany) equipped with a 7 T refrigerated, actively shielded superconducting magnet (Bruker Biospin, Wissembourg, France). The samples were prepared at a concentration of 1.0 mg mL^−1^ in THF, while the matrix (DCTB) was mixed at a concentration of 10.0 mg mL^−1^.

The glass transition temperature (T_g_) of the polymers was measured by differential scanning calorimetry (DSC) using a DSC 2920 apparatus manufactured by TA Instruments under a nitrogen flux of 50 mL min^−1^ with a heating and cooling rate of 10 °C min^−1^ in the range −10 to 200 °C. All calorimetric data were reported for the second heating cycle [65].

TGA analysis was performed with a Thermogravimetric analyzer (TGA) from TA Instruments: TGA Q500, Netzsch: TG 209 F1.

## 4. Conclusions

Salen-type chromium complexes supported by salen, salalen, and salan ligands with t-Bu substituents on the phenolate moieties and an ethylene bridge between the nitrogen donor atoms were synthesized and fully characterized. These air-stable complexes were then tested as catalysts to produce polycarbonates and polyesters by ring-opening copolymerization of epoxides with CO_2_ and phthalic anhydride, respectively. More flexible salalen and salan complexes resulted in greater activity than the salen chromium complex in the reaction of CO_2_ with cyclohexene oxide. With all complexes, polycyclohexene carbonates were produced exclusively. *T*_g_ and *T*_d_ temperatures determined by DSC and TGA, respectively, were in line with the values reported in the literature for analogous PCHC samples.

All complexes revealed high activities and selectivities in the copolymerization of cyclohexene oxide and limonene oxide with phthalic anhydride to produce polyesters with perfectly alternated structures. For these reactions, the salen complex showed the highest activity, reaching values comparable with those of the commercial salen complex.

Finally, all catalysts were able to promote switch catalysis between these two processes for the synthesis of diblock polyester/polycarbonate copolymers.

Interestingly, the synthesized chromium complexes proved to be excellent candidates also for the depolymerization of polycyclohexene carbonate. Salen complex **1** was the most efficient catalyst, allowing the almost quantitative chemical recycling of the cyclohexene oxide monomer. The same complex was also able to promote the depolymerization of the polycarbonate segment of the diblock copolymer (albeit at lower rates), suggesting that the chemical recycling of CHO is possible even when the PCHC is incorporated in more complex polymeric structures.

These results demonstrated that salen chromium complexes offer new opportunities to develop efficient catalysts both for the synthesis and the chemical degradation of polycarbonates. Thus, they could have a key role in implementing a circular economic model for plastic materials, with benefits including reduced pressure on the environment, enhanced raw materials, increased competitiveness, and innovation.

## Data Availability

Not applicable.

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
