# Peer review of "Chromium Complexes Supported by Salen-Type Ligands for the Synthesis of Polyesters, Polycarbonates, and Their Copolymers through Chemoselective Catalysis"

_ijms, 2023, doi:10.3390/ijms24087642_

Round 1

Reviewer 1 Report

Comments to Authors

1.      Write keywords in alphabetical order. Remove switch catalysis de-polymerization.

2.      Add the impact of current work on industry and future research.

3.      Authors need to improve the problem statement in the introduction section.

4.      Why and how the said parameters were selected for this work? More specific details needed

to be added with the use of the latest reference.

5.      Revise the last paragraph of the introduction section.

6.      Explain Figure-2 in more detail.

7.      3. Materials and Methods. Revise it and make it clear.

8.      Electronic absorption spectra. Better to put it into the manuscript.

9.      GPC, TGA, and DSC mentioned some references with each graph.

10.   In your discussion section, please link your empirical results with a broader and deeper literature review.

11.   Explain the conclusion in more detail.

Authors need to increase the literature and problem statement from the current recent papers such as.

v  https://doi.org/10.3390/polym13020268 

v  https://doi.org/10.1002/app.51191

v  https://doi.org/10.1016/j.envres.2023.115253

Author Response

The authors thank the reviewer for his/her suggestions that contribute to improve the quality of the paper.  All modifications in the revised version of the paper  are highlighted in yellow.

  1. Write keywords in alphabetical order. Remove switch catalysis de-polymerization.

Response: The keywords have been modified as suggested. Polycarbonate was added as additional keyword.

  1. Add the impact of current work on industry and future research.

Response: As suggested by the reviewer, a sentence about the impact of current work on industry and future research was added in the conclusions.

  1. Authors need to improve the problem statement in the introduction section.

Response: As suggested by the reviewer, this point has been  better addressed in the last paragraph of the introduction.

  1. Why and how the said parameters were selected for this work? More specific details needed

Response: As suggested by the reviewer, some additional information about the synthesis of the catalysts and the polymerization conditions  were added in the paper.

  1. Revise the last paragraph of the introduction section.

Response: The last paragraph was partially rewritten to make it clearer. The modifications are highlighted in yellow.

  1. Explain Figure-2 in more detail.

 Response: The descriptions of the mechanisms reported in Fig 2  were  added in the paper 

  1. 3. Materials and Methods. Revise it and make it clear.

Response: Some additional information about the methods used have been added.

  1. Electronic absorption spectra. Better to put it into the manuscript.

Response: As suggested by the reviewer, the spectra were added in the paper as Figure 1

  1. GPC, TGA, and DSC mentioned some references with each graph.

Response: Details about instruments and techniques (GPC, TGA, and DSC analysis) were added in the paper.

  1. In your discussion section, please link your empirical results with a broader and deeper literature review.

Response: As suggested by the reviewer, some comparisons with results already reported in the literature have been added.

  1. Explain the conclusion in more detail.

Response: The conclusion was modified to be more detailed.

Authors need to increase the literature and problem statement from the current recent papers such as.

v  https://doi.org/10.3390/polym13020268 

v  https://doi.org/10.1002/app.51191

v  https://doi.org/10.1016/j.envres.2023.115253

Response: The first reference was added as requested  by the reviewer  

Reviewer 2 Report

The reviewed work needs a number of editorial corrections, such as the title of the manuscript ends with a period or the lack of paragraphs. There are many such moments, where these minor, but at the same time glaring editorial errors should be eliminated.  The present work lacks a detailed discussion of reaction mechanisms. The authors of the publication cite the conversion function, while the text lacks any figures on the basis of which they drew such and not other conclusions. Also missing is the formula on the basis of which such a function was calculated. All the experiments carried out by the authors are based only on dry conclusions unsupported by any numerical values, which definitely reduces the value of the reviewed work and which should be absolutely supplemented. 

Author Response

The reviewed work needs a number of editorial corrections, such as the title of the manuscript ends with a period or the lack of paragraphs. There are many such moments, where these minor, but at the same time glaring editorial errors should be eliminated. 

The present work lacks a detailed discussion of reaction mechanisms. The authors of the publication cite the conversion function, while the text lacks any figures on the basis of which they drew such and not other conclusions. Also missing is the formula on the basis of which such a function was calculated.

All the experiments carried out by the authors are based only on dry conclusions unsupported by any numerical values, which definitely reduces the value of the reviewed work and which should be absolutely supplemented. 

 The authors thank the reviewer for his/her suggestions that contribute to improve the quality of the paper.  All modifications in the revised version of the paper  are highlighted in yellow. All modifications performed are listed below:

1. The paper was carefully checked to remove all the typos and the editorials mistakes.

2. The title of paragraph 2.1 was modified.

3. The mechanisms of the two ROCOP reactions were added in the related paragraphs.

4. The calculations of conversions were added in the captions of the Tables 1 and 2.

Round 2

Reviewer 2 Report

The authors responded to the reviewer's comments in the revised paper. A minor error that catches the eye is:

Fig. 2. 2D DOSY NMR (600 MHz, CDCl3, 298 K) of poly(cyclohexene phthalate-cyclohexene carbonate) (entry 2 in Table 3) - the graph should be m2/s, where "2" in superscript.

Other than that, there are no more comments on the reviewed portion.